# Late Miocene Tarim desert wetting linked with eccentricity minimum and East Asian monsoon weakening

Junsheng Nie [1,2✉], Weihang Wang [1], Richard Heermance[3], Peng Gao[1], Li Xing[1], Xiaojian Zhang [4✉], Ran Zhang [5✉], Carmala Garzione[6] & Wenjiao Xiao [7✉]

Periodic wetting is an inherent feature of many monsoon marginal region deserts. Previous studies consistently demonstrate desert wetting during times of Earth's high orbital eccentricity and strong summer monsoon. Here we report the first evidence demonstrating desert wetting during Earth's low orbital eccentricity from the late Miocene strata of the northwestern Tarim Basin of northern China, which is commonly thought to be beyond the range of Asian monsoon precipitation. Using mechanisms for modern Tarim wetting as analogs, we propose that East Asian summer monsoon weakening enhanced westward moisture transport and caused opposite desert wetting pattern to that observed in monsoon marginal region deserts. This inference is supported by our model simulations. This result has far-reaching implications for understanding environmental variations in non-monsoonal deserts in the next few thousands of years under high atmospheric $CO_2$ content and low eccentricity.

[1] Key Laboratory of Western Chinese Environmental System (Ministry of Education), College of Earth and Environmental Science, Lanzhou University, Lanzhou, China. [2] Center for Excellence in Tibetan Plateau Earth Sciences, Chinese Academy of Sciences, Beijing, China. [3] Department of Geological Sciences, California State University, Northridge, CA, USA. [4] School of Geography and Ocean Science, Nanjing University, Nanjing, China. [5] Institute of Atmospheric Physics, Chinese Academy of Sciences, Beijing, China. [6] Department of Environmental Sciences, Rochester Institute of Technology, Rochester, NY, USA. [7] Xinjiang Research Center for Mineral Resources, Xinjiang Institute of Ecology and Geography, Chinese Academy of Sciences, Urumqi, China. ✉email: jnie@lzu.edu.cn; zhangxj@nju.edu.cn; zhangran@mail.iap.ac.cn; wj-xiao@mail.iggcas.ac.cn

One of the most exciting findings in terrestrial paleoen-vironmental research is that deserts on the margin of monsoon climate regions experience cyclic vegetation coverage (greening) and wetting in the geological past[1–6]. Desert wetting is thought to have provided a green corridor for human migration out of Africa[5] and may also have relieved water stress for nomadic civilization and decreased war potential between the nomadic and agricultural civilizations[7]. Therefore, understanding desert greening and wetting patterns and the underlying reasons are important scientific questions of wide interests.

For example, many studies have been done on the Sahara Desert, and results reveal that Saharan greening is tied to African summer monsoon intensification driven by insolation variations and vegetation, dust, and soil feedbacks[6,8]. Furthermore, longer sedimentary records reveal periodic Saharan greening associated with high eccentricity values[1], suggesting insolation forcing of Saharan environmental variations.

Similar to the Saharan desert records, work in the northern China deserts reveals that dune stabilization and soil development occurred a couple of thousands of years (kyr) after peak Northern Hemisphere (NH) summer insolation during the Holocene, leaving room for vegetation, dust, and soil feedbacks and NH ice sheet forcing[3,4,9,10]. This Holocene dry-wet pattern inferred from the northern China desert records is consistent with East Asian summer monsoon (EASM) variations established from the Chinese Loess Plateau (CLP) records[2,11], with dune stabilization corresponding to a strong EASM. Most paleoenvironmental records from northern China deserts covering the past several glacial-interglacial cycles do not have good age control, but the neighboring CLP records have relatively good age constraints, and they show that EASM was stronger during high eccentricity[12].

In comparison with the monsoonal region deserts, little is known about orbital timescale environmental variations in non-monsoonal region deserts. Instead, most of the previous work on deserts in non-monsoonal regions has focused on their formation ages[13–17]. A recent study suggests that the Taklimakan desert of the Tarim Basin of northern China has been in perennial desert conditions since 0.7–0.5 Ma with no apparent greening since that time[17]. However, with >400 ppmv atmospheric $CO_2$ levels and a continued warming climate, the future Taklimakan desert may not exhibit similar features to the late Quaternary, which had $CO_2$ level of ca. 180–280 ppmv[18]. In order to identify a better analog to future climate, one needs to go back to beyond the Quaternary. Heermance et al.[16] reported the existence of over 1000 meters of alternating dune and interdune strata between 12 and 7 Ma within the northwestern Tarim Basin, which provides a good opportunity to assess whether non-monsoonal region deserts experienced periodic greening and wetting in higher $CO_2$ conditions.

Here we show that the late Miocene Tarim desert experienced periodic wetting and greening in eccentricity minimum and EASM weakening.

The Tarim Basin is bounded by the Tianshan, Pamirs, and the Kunlun ranges to the north, west, and south, respectively (Fig. 1). This basin receives moisture mostly from westerlies, although during past warm interglacials, speleothem oxygen isotope data suggest that Asian summer monsoon could penetrate to this area[19]. The study site is located in the northwestern corner of the Tarim Basin in the rain shadow zone of the Pamir and Tianshan Mts. (Fig. 1). Uplift of the Pamir and the Tianshan Mts. has been a prolonged process with phases of uplift during the Oligocene and Miocene, although stable isotope data indicate that not until 12 Ma did the Pamir and Tianshan form a topographic barrier that blocked moisture from the atmospheric westerlies[16,20,21].

The studied West Kepintagh section is 3800 m thick, including Wuqia group, Atushi Formation, and Xiyu Formation, and has been dated by magnetostratigraphy (~17.5–1.5 Ma). In total,

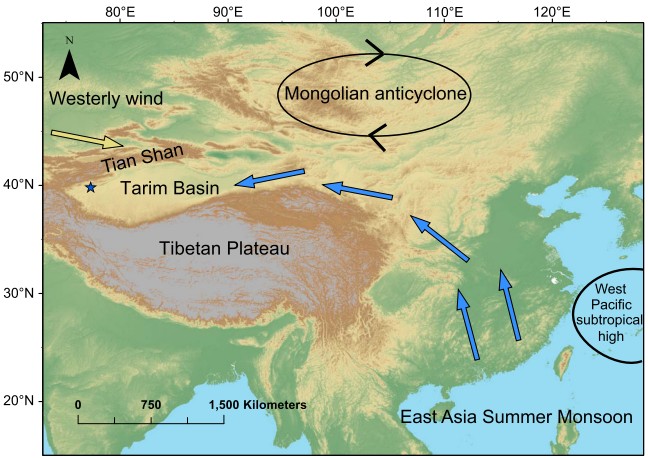

**Fig. 1 Schematic map showing increased moisture transport to the Tarim Basin during weak East Asian summer monsoon circulation.** We note that climate data from the past few decades suggest that the westward shift of the West Pacific subtropical high and intensified Mongolian anticyclone associated with East Asian summer monsoon weakening promoted westward moisture transport[41], which is shown by the blue arrows. Star shows the location of the study site.

1100 m eolian dune-containing strata occur in the upper Wuqia group and they span from 12.2 to 7.0 Ma according to the magnetostratigraphy[16]. In order to study desert wetting-drying cycles recorded by this section, we performed new fieldwork and obtained high-resolution samples from the upper 725 m strata of the eolian portion. The other portions are not accessible due to earthquake-caused landslides. Most of the studied interval is dominated by eolian sandstone, with interdune siltstone only taking ~20% of the thickness, except the bottom 50 m (Fig. 2). This suggests an interval of dry environment with occasional wetting. By contrast, the bottom 50 m is dominated by interdune siltstone, taking up 60% of the 50 m thickness (Fig. 2), suggesting a wet environment-dominated interval. Indeed, the unsampled underlying eolian dune-containing strata are also dominated by interdune siltstone[16] (Supplementary Fig. 1), consistent with the bottom 50 m of the sampled strata, suggesting intensified aridi-fication for the interval above 50 m location of this study.

## Results

**Age model establishment**. Six full normal and one partial normal, and six full and one partial reversed polarities are observed (Fig. 2). Under the age model framework[16] of Heermance et al., which is consistent with suggested ages of the geological map in this area[22], the six full normal and reversed polarities can be correlated to C4n.1r-C5n.1n (Fig. 2), with top and bottom age of 7.642 Ma (top C4n.1r) and 9.937 Ma (bottom C5n.1n), respectively. Using an average sediment accumulation rate of 0.31 m kyr$^{-1}$ for this interval covering C4n.1r-C5n.1n, we estimated the duration of 13 geomagnetic reversals as recorded by samples in this section (Supplementary Table 1). Eight of 13 reversals of the studied interval are constrained by samples of opposite polarities that are <20 kyr apart (marked with red triangles in Fig. 2); two reversals are constrained by samples of opposite polarities that are between 35 and 40 kyr apart (marked with purple triangles in Fig. 2). However, the other three reversals are loosely constrained, with neighboring samples exhibiting opposite polarities 75, 78, and 125 kyr apart (marked with blue triangles in Fig. 2). Assigning middle meters between samples recording opposite polarities to their corre-sponding reversal ages in Geologic Polarity Timescale[23], we established the age model for the section based on piecewise

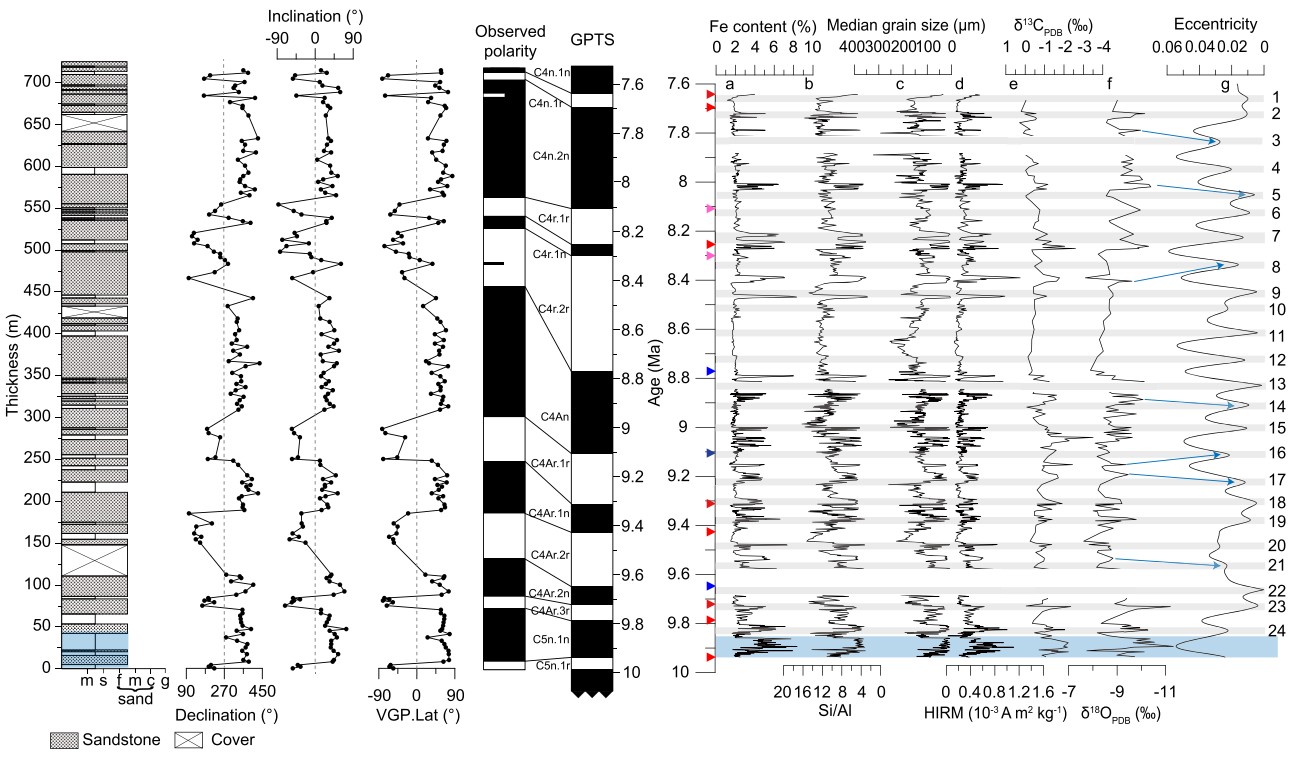

**Fig. 2 Lithology, age model, and paleoenvironmental proxy records from the studied section.** The left panel shows the lithology and paleomagnetic age model. The right panel shows paleoenvironmental proxy records. **a** Fe content. **b** Si/Al. **c** Median grain size. **d** HIRM. **e** $\delta^{13}C_{PDB}$. **f** $\delta^{18}O_{PDB}$. **g** Eccentricity[24]. Gray bars show the disappearance of eolian dune strata during low eccentricity. The blue shadow highlight dominant wet environment over the lower ~50 m of the strata. The triangles show reversals recorded by this section, with red, purple, and blue ones corresponding to tightly (<20 kyr), mediumly (35–40 kyr), and loosely (75–125 kyr) constrained reversals, respectively. Numbers on the right indicate eccentricity lows from top to bottom.

interpolation (Fig. 2). We note that the three reversals marked by blue triangles have large uncertainties, which propagate to the portion of strata bounding by them in the process of piecewise interpolation. There are three covered intervals in the studied section (~10% in thickness), but the upper two covered intervals do not align with geomagnetic reversals (Fig. 2), so their existence does not increase uncertainties of our age model. The lower covered interval does overlap with one geomagnetic reversal (Fig. 2), but there are five tightly constrained reversals (<20 kyr) from 10–9.3 Ma for the lower part of the section, so the existence of this covered interval only has limited impact on the precision of our age model.

**Paleoenvironmental reconstruction and simulations**. We performed grain size, Fe%, Si/Al, carbonate oxygen and carbon isotope ratio, and magnetic parameter analyses (such as hard isothermal remanent magnetization (HIRM)) on the samples (Fig. 2 and Supplementary Fig. 2). These parameters exhibit similar variations, with smaller median grain size of interdune lacustrine/fluvial overbank strata corresponding to higher Fe%, HIRM values, lower Si/Al ratios, and more negative oxygen and carbon isotope ratios (Fig. 2 and Supplementary Figs. 3 and 4). Interestingly, most interbedded interdune siltstone layers of the sequences correspond with low eccentricity[24] for Earth's orbital parameters (i.e., more circular orbit; Fig. 2). Some slightly misaligned siltstone layers (highlighted by blue arrows)[25] corresponds to strata constrained by reversals with large uncertainties. By contrast, for those located near reversals or between two short reversals with smaller uncertainties (red and purple triangles), the alignments are more robust. Particularly, two siltstone layers (near 9.3 and 9.7 Ma) occur right at two reversals, and they align perfectly with eccentricity minima (Fig. 2), supporting our

interpretation that the slightly misaligned siltstone layers to eccentricity minima are due to age model uncertainties. Exceptions occur for the siltstone-dominated lower 50 m, where sandstone aligns with low eccentricity (Fig. 2, blue area), suggesting a different pattern from the upper sandstone-dominated strata (>50 m). We note above that the unsampled eolian dune-contained strata below the studied interval are siltstone dominated, similar to the bottom 50 m of the studied section. We interpret the siltstone-low eccentricity alignment pattern from 50 m upward may not be applied to the lower sequence, which was dominated by wetter environments.

We simulated the effects of eccentricity variations on summer precipitation in the Tarim Basin and the monsoonal region (Fig. 3). Our results indicate that eccentricity decrease results in contrasting precipitation patterns between the Tarim Basin and monsoonal regions, such as the CLP and the northeastern Tibetan Plateau. Low eccentricity corresponds to increased precipitation in the Tarim Basin but decreased precipitation in the monsoonal region (Fig. 3).

## Discussion

**Tarim wetting linked with low eccentricity for dune-dominated strata.** Fe%, and Si/Al ratio are useful parameters to recognize wet-dry variations in desert region paleoenvironmental studies because eolian sand strata have high Si and low Fe content[26]. The inverse correlation between these parameters with medium grain size (higher values corresponding to dune strata) confirms this point. Following this correlation, we interpret higher hematite content (inferred by higher HIRM values) as a result of increased chemical weathering associated with Tarim wetting, which provided moisture for hematite formation. This is consistent with the

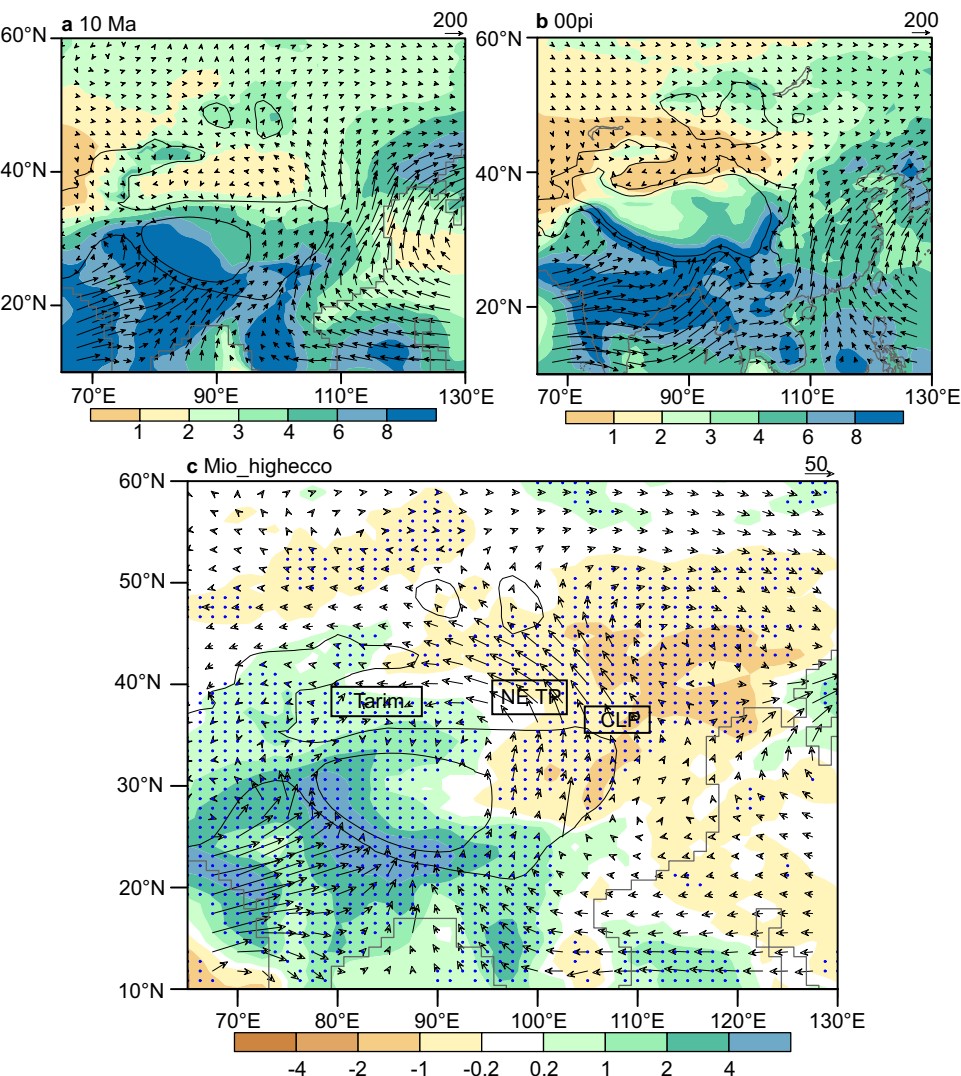

**Fig. 3 Simulated summer (JJA) precipitation (shaded, mm day⁻¹) and water vapor transport integrated from the surface to 700 hPa (vector, units: kg m⁻¹ s⁻¹).** JJA June, July and August. **a** Results using 10 Ma background boundary conditions and orbital parameters for 1950. **b** Results using pre-industrial background boundaries and orbital parameters for 1950. **c** Precipitation and water vapor transport differences when increasing eccentricity from 0.01672 (year 1950) to 0.05. The difference is calculated by subtracting the high eccentricity results from the low eccentricity results. Green (yellow) color indicates more precipitation under lower (higher) eccentricity. Topography levels equal to 1500 m and 3000 m are highlighted with gray contours. The three black rectangles in **c** represent the rough locations of the Tarim Basin, Northeastern (NE) Tibetan Plateau (TP) and the Chinese Loess Plateau (CLP). Dotted regions indicate differences that are significant at the 90% confidence level.

pattern on the CLP, where paleosol and loess layers have high and low HIRM values, respectively, interpreted as a result of intensified weathering and pedogenesis over soil formation processes[27,28]. Anhysteretic remanent magnetization (ARM) susceptibility data from the studied section covary with HIRM (Supplementary Fig. 2), suggesting that hematite production in the Tarim Basin over the late Miocene was accompanied with production of nanometer-scale ferrimagnetic grains (proportional to ARM susceptibility), consistent with the pedogenic model of Chinese loess[28,29].

In order to examine whether the wetting intervals are accompanied with desert greening, we counted charcoal content in two fluvio-lacustrine and three eolian sandstone samples. We found that the eolian sandstone samples have low charcoal content (120, 298, and 1054 grains/g, Supplementary Fig. 5); in contrast, the fluvio-lacustrine samples have high charcoal content (4016 and 8056 grains/g, Supplementary Fig. 5). These results not only confirm climate wetting corresponding to fluvio-lacustrine

strata, but also suggest desert greening accompanied with climate wetting. Stable isotope data also provide support to the above interpretation because siltstone occurrence aligns with more negative oxygen and carbon isotope values (Fig. 2 and Supplementary Fig. 4), which indicates more water availability and better vegetation cover[30–32]. Carbonates formed under wetter conditions and better vegetation cover have more negative $\delta^{13}C$ values due to increased soil-respired $CO_2$ contribution over precipitation of carbonates[33], as well as more negative $\delta^{18}O$ values due to increased contribution of soil water which has more negative $\delta^{18}O$ associated with more precipitation. On the other hand, in conditions with little vegetation cover and less precipitation, carbonates would have less negative $\delta^{13}C$ values associated with decreased respired $CO_2$ contribution, as well as less negative $\delta^{18}O$ values associated with less precipitation. One complication to the above interpretation is that carbonates in these strata may be detrital, but examining thin sections reveals that this is not the case. For sandstone, detrital components are

mostly quartz (>80%) and feldspar (15%); carbonates are micritic taking up the pore spaces (Supplementary Fig. 6a, b). Siltstone has carbonate as cement and pelitic texture (Supplementary Fig. 6c, d). The pelitic texture contains about 70% microcrystalline clay minerals, 30% quartz, and few micas. Detrital carbonate fragments are not observed in the thin sections of these two samples. Detrital carbonate interpretation for siltstone intervals also contradicts with increased HIRM and ARM susceptibility values, suggesting occurrence of some degree of soil formation or chemical weathering (Fig. 2 and Supplementary Fig. 2).

The interval of 8.8–8.5 Ma has three eccentricity lows but our paleoenvironmental data do not show clear changes. We notice that this interval has low sediment accumulation rates (Supplementary Fig. 3). We suspect that our equal interval sampling strategy may have skipped the interdune layers because interdune layers may be thinner over this interval than the intervals with high sediment accumulation rates. Alternatively, this interval may include hiatuses caused by erosion of sand dunes, which have removed interdune layers. Despite these possibilities, a careful examination of the grain size data (Fig. 2 and Supplementary Figs. 7 and 8) shows three intervals of grain size decrease and silt proportion increase between 8.8 and 8.5 Ma, aligning well with low eccentricity. Furthermore, content of nanometer-scale ferrimagnetic grains and hematite increased over the low eccentricity intervals of 8.8–8.5 Ma, as can be told from increased ARM susceptibility and HIRM values (Supplementary Fig. 7), indicating enhanced weathering associated with environmental wetting. This pattern is consistent with the observed wetting aligning with low eccentricity pattern in the other intervals.

In summary, the consistency of the paleoenvironmental proxy datasets (geochemistry, stable isotopes, magnetic mineralogy, and vegetation data) with model simulations is best interpreted through invoking environmental changes instead of factors like preservation or sedimentary sorting bias.

**Forcing for late Miocene Tarim wetting**. The observed ancestral Taklimakan desert wetting and greening during eccentricity minima over dune-dominated strata could, in theory, come from increased precipitation from westerly atmospheric flow[32] or by the westward expansion of the Asian monsoon[19]. However, we suggest that neither mechanism is feasible for causing the observed wetting. First, formation of the ancestral Taklimakan desert at ~12 Ma is attributed to uplift of the Pamir and the Tianshan Mts., which formed a rain shadow region to the studied area and blocked the moisture from the westerlies[16] (Fig. 1). Thus, westerly moisture sources would have been cut off to this region. Second, many records show weakened monsoon precipitation during Earth's orbital eccentricity minima (Supplementary Fig. 9c) associated with decreased sea-land pressure gradient during this orbital configuration[34–39].

Precipitation in the Tarim Basin increased in the recent decades under the background of EASM weakening[40,41]. Thus, understanding forcing mechanisms for recent precipitation increase in the Tarim Basin may shed light on its wetting mechanism during the late Miocene because these two periods have similar $CO_2$ levels. Chen et al. found that increased precipitation in the Tarim Basin during the past 50 years was mainly caused by the weakened EASM[41]. A weakened EASM was associated with increased frequency of anticyclone activity in Mongolia, which caused increased frequency of moisture-laden easterly winds that transport monsoonal moisture into the Tarim Basin along the northern periphery of the Tibetan Plateau (Supplementary Fig. 10). Although this moisture transport did not occur year round, it typically brought heavy precipitation to this area during summer months (Supplementary Fig. 11)[40], so

that even days of such moisture transport would have impacted the total precipitation in the Tarim Basin[40].

In order to test whether monsoonal moisture transport can also explain ancestral Taklimakan Desert wetting during weakened EASM, we performed climate simulations using the Community Atmosphere Model version 4 (CAM4) (Supplementary Table 2) based on 10 Ma boundary conditions[42]. Model results show a prevailing westward monsoonal moisture transport toward the Tarim Basin along the northern periphery of the Tibetan Plateau throughout the summers during the late Miocene, which is associated with anticyclonic circulation over Mongolia (Fig. 3a). However, such monsoonal moisture transport only exists at synoptic timescales in summers at present (Supplementary Fig. 10b) due to the prevailing westerlies over the northern periphery of the Tibetan Plateau (Fig. 3b and Supplementary Fig. 10a). Therefore, monsoonal moisture was likely more easily transported to the study area during the late Miocene than that during the recent decades. Model results further demonstrate that both summer and annual precipitation in the Tarim Basin and surrounding mountains under low eccentricity increased in comparison with under high eccentricity values (Fig. 3 and Supplementary Fig. 12). The increased precipitation was also accompanied with enhanced westward monsoonal moisture transport. We therefore propose that the mechanism for the modern Tarim wetting under weakened EASM precipitation can likely explain ancestral Taklimakan Desert wetting during weakened EASM, although the driving forces of the EASM may differ between the orbital and decadal timescales[43–45]. In contrast, summer precipitation in the East Asian monsoonal region, such as the CLP and Northeastern Tibetan Plateau area (Fig. 3c), decreased under low eccentricity, consistent with what the geological data from these regions suggest[46–48] (Supplementary Fig. 9c). These features are consistent with our predictions. We note that annual mean precipitation also shows similar pattern as the summer mean precipitation (Supplementary Fig. 12c), but the magnitude contrast between high and low eccentricity is less obvious than that in the summer precipitation case (Fig. 3c). This is consistent with our hypothesis because our model specifically emphasizes importance of summer precipitation.

Our model simulations do not show significant change of precipitation in the Tarim Basin upon changing eccentricity, when the EASM was strong, like 6 ka[2–4] (Supplementary Fig. 13). This is consistent with our hypothesis because monsoonal moisture has been precipitated in East Asia and limited moisture is available in East Asia to be transported to the Tarim Basin. Therefore, when the EASM was strong, Tarim precipitation would have low sensitivity to eccentricity variations. If our model is correct, late Miocene EASM records should have clear eccentricity-band signal, several such records are available for the late Miocene, and strong eccentricity-band signals do appear for records with enough resolution[46,49]. Particularly, Nie et al. show high precipitation in the Qaidam Basin[46] was aligned with high eccentricity over the late Miocene (Supplementary Fig. 9c), supporting our model which requires weak EASM coupled with high Tarim precipitation.

We also explore the sensitivity of northern China precipitation variations to precessional forcing. In the low eccentricity setting (Supplementary Fig. 14a), the precipitation variations in the Tarim Basin and the NE Tibetan Plateau/CLP do not show sensitive responses to precessional variations. This is reasonable because the forcing (insolation) has lower amplitude when eccentricity is low.

In the high eccentricity setting (Supplementary Fig. 14b), the precipitation variations in the Tarim Basin and eastern China (referred here to roughly 25–50° N; 95–120° E; including the NE Tibetan Plateau/CLP) show sensitive responses to precessional

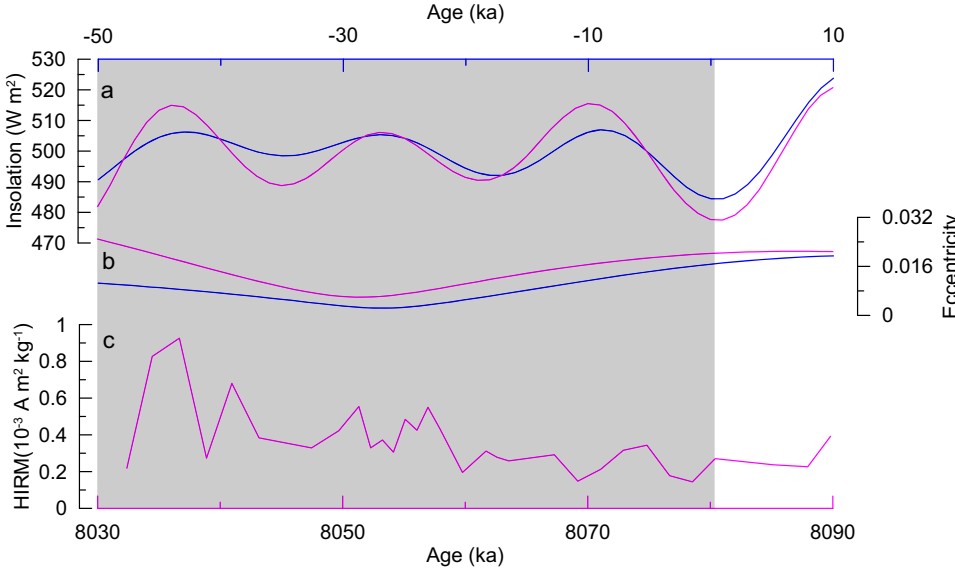

**Fig. 4 Summer insolation and eccentricity over 8.09–8.03 Ma compared with present and future configuration and the HIRM time series. a** Summer insolation comparison between the intervals of 8.03–8.09 Ma (purple) and from 10 ka to future 50 kyr (blue). Both records are June 21 insolation from 40°N[24]. **b** Eccentricity comparison between the intervals of 8.09–8.03 Ma and from 10 ka to future 50 kyr[24]. **c** HIRM from Tarim over 8.09–8.03 Ma. We used the tuned age model for the Tarim record, with tuning tie points shown by blue arrows in Fig. 2. The gray bar highlights the interval of low amplitude insolation variations for the next 50 kyr, similar to those during 8.08–8.03 Ma.

variations, and these two areas show opposite precipitation pattern, just like that in Fig. 3c. It also explains why the Tarim Basin was wetter in low eccentricity intervals than high eccentricity intervals (Fig. 2), which can be attributed to moisture penetration to the Tarim basin only at a portion of the high eccentricity intervals controlled by precessional variations. In contrast, under low eccentricity setting (corresponding to less eastern China precipitation associated with weaker EASM), moisture could have been more commonly transported to the Tarim Basin because Tarim precipitation was not sensitive to precessional value variations in this eccentricity setting.

We note that our simulations show a weaker EASM precipitation in eastern China under 6 ka precession setting than that under present precession setting, with the late Miocene boundary conditions. This is consistent with the results of Marzocchi et al.[50]. And Dai et al. attributed decreased precipitation in eastern China to the westward shift of the western Pacific Subtropical High[51], which was also detected in our simulation under 6 ka precession setting, shown by anomalous anticyclonic moisture transport in eastern China (Supplementary Fig. 14b). These studies are consistent and complimentary to our simulations.

In summary, we present the first demonstration that non-monsoonal regional deserts would experience periodic wetting and greening in a high $CO_2$ and low eccentricity world without NH permanent ice sheets[52–54], a possible analog to the near future. However, different from the monsoonal region deserts, non-monsoonal region deserts show an opposite wetting pattern to the monsoonal region deserts and summer monsoon intensity[1,4,36]. Although eccentricity minimum and Asian monsoon weakening may not be the exclusive reason for Tarim desert wetting and greening during the late Miocene, the data suggest that this is the major mechanism.

The orbital setting of Earth in the next 50 kyr is similar to that of 8.08–8.03 Ma, with a new eccentricity minimum in ~30 kyr. Interestingly, the paleoclimatic records reported here for 8.08–8.03 Ma show that climate in the Tarim Basin was wetter during times of eccentricity minima (Fig. 4). If the late Miocene

can be used as an analog for future climate, then the Tarim Basin would become wetter over the next 30 kyr.

## Methods

**Paleoclimate reconstruction.** In total, 230 paleomagnetic samples were oriented with a Brunton compass and were taken at an interval of 0.5–2 m, using a portable gasoline-powered drill. They were then exposed to 17 steps of thermal demagnetization from room temperature to 680 °C, using a MMTD-80 thermal demagnetizer, followed by measuring of remanent magnetism using a 2G Superconducting Rock Magnetometer after each step of thermal demagnetization.

Before environmental magnetic analysis, samples were first dried at 35 °C in an oven. Then about 10 g samples were crushed and packed into a plastic box of 2 cm × 2 cm × 2 cm. Isothermal remanent magnetism was imparted first at 1.2 T and then at −0.3 T, using an ASC IM-100 impulse magnetizer, followed by remanence measurements using a JR-6A spinner magnetometer. The HIRM is calculated as the sum of $IRM_{1.2T}$ and $IRM_{-0.3T}$ divided by 2. The ARM was imparted with ASC D2000 alternating-field demagnetizer with a peak alternating-field of 100 mT and a direct current field of 0.05 mT, then measured with a JR-6A spinner magnetometer. ARM is often expressed by its susceptibility format by normalizing by the direct current field. All the analyses were undertaken at the Paleomagnetism and Environmental Magnetism Laboratory, China University of Geosciences. Low-frequency (475 Hz) magnetic susceptibility was measured in Bartington MS2 at Key Laboratory of Western China's Environment Systems (Ministry of Education), Lanzhou University, China.

For grain size analysis, disaggregated samples were analyzed using a Malvern Mastersizer 2000 particle size analyzer. Prior to analysis, samples were pretreated with hot hydrogen peroxide to remove organic matter and then with hydrochloric acid to remove carbonate, following the standard procedure at Lanzhou University.

The X-ray fluorescence analyses were carried out on powdered samples using a Rigaku D/Max-2400 diffractometer with a Cu Ka (l = 1.54056 Å) radiation source operating at 40-kV voltage and 60-mA current in the Key Laboratory of Western Chinese Environmental System (Ministry of Education), Lanzhou University. The X-ray diffraction patterns were obtained at a scanning rate of 15°/min, with a step width in the 2q scan of 0.02° in the range 3° to 80°. The intensity of each mineral was estimated from peak height.

Carbon and oxygen isotopes were measured on a Delta V Advantage gas source isotope mass spectrometer in the Linyi University, China, and the results are expressed using the δ notation relative to the PDB standard (with a precision of less than 0.1‰). Before measurement, samples were oven-dried at 40 °C and ground to power, followed by reacting with 100% phosphoric acid at 90 °C in Gasbench to release $CO_2$ gas.

**Model simulations.** In this study, we use the CAM4 with a horizontal resolution of ~1°, configured by ~0.9° in latitude and 1.25° in longitude with 26 vertical layers.

At this resolution, CAM4 uses a finite-volume dynamical core. Version 4 of the Community Land Model (CLM4) is also included[55] and shares the same horizontal resolution as CAM4. Generally, CAM4 reasonably reproduces the large-scale pattern of modern Asian climate[56] and has a good ability to investigate Asian paleoclimatic change[57].

We conduct CAM4 experiments based on ~10 Ma boundary conditions. Due to the long simulation time required by using a high-resolution fully coupled model, we use the simulated ~10 Ma sea surface temperatures (SSTs) to force CAM4. These SSTs are from a ~10 Ma experiment using the low-resolution Norwegian Earth System Model (NorESM-L) (atmosphere, ~3.75°)[42] under ~10 Ma paleogeography[42,58] with 350 ppmv of atmospheric $CO_2$ concentration and Antarctic ice sheet, with ice sheet size and topography derived from late Pliocene conditions set by the Pliocene Model Intercomparison Project[59].

We conduct two CAM4 experiments in total (Supplementary Table 2). First, we run a CAM4 experiment to simulate the ~10 Ma atmospheric conditions (named Mio) using the ~10 Ma paleogeography[42,58], 350 ppmv of atmospheric $CO_2$ concentration, an Antarctic ice sheet[59], modern orbital parameters (1950) and associated SSTs simulated by NorESM-L[42]. Based on this experiment, we conduct another experiment (named Mio_highecco) by changing the eccentricity from 0.01672 (Mio) to 0.05. Comparing these two experiments, we can check the climatic effects of decreased eccentricity under the ~10 Ma paleogeography. Furthermore, we conducted two more experiments (Mio6k and Mio6k_highecco). These two experiments used the same boundary conditions as these in experiments Mio and Mio_highecco, except using the precession at 6 ka. With these two experiments, we can further check the effects of changing precession on precipitation in the Tarim Basins. Each CAM4 experiment is run for 25 model years. These experiments reach a quasi-equilibrium state in their first 5 model years; therefore, the computed climatological means of the last 20 model years are analyzed here.

## Data availability
Source data are provided with this paper.

## Code availability
The model simulation code used to support our dynamic interpretations is available at: https://zenodo.org/record/6215542.

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

## Acknowledgements

This work was financially supported by the Second Tibetan Plateau Scientific Expedition (Grant 2019QZKK0704), the National Natural Science Foundation of China (Grants 41888101 and 42030505), Science and Technology Department of Gansu Province, China (Grant 20JR5RA260), the United States National Science Foundation (Grants EAR-1348005, EAR-134-8075, and OISE-1545859), and American Chemical Society Petroleum Research Fund grant (50776-UN18). We thank Pu Li, Zeng Luo, and Fangbin Liu for collecting and analyzing the samples, Wenxia Han for stable isotope analysis help, and Yifan Hua, Xiaofeng Ma, Dan Breecker, Feng Cheng, Lin Li, and Xiangzhong Li for preparing and/or discussing thin sections.

## Author contributions

J.N., R.H., X.Z., C.G., and W.X. designed research. W.W., P.G., L.X., and R.Z. performed research. All authors discussed the data. J.N. wrote the manuscript with the help of other coauthors.

## Competing interests

The authors declare no competing interests.
