## [Peer Review File · Nature Communications]

Late Miocene Tarim desert wetting linked with eccentricity minimum and East Asian monsoon weakeningReviewers' Comments:

Reviewer #1:

Remarks to the Author:

General comment:

This manuscript (MS) aims to discuss the orbital forcing of dry-wet cycle in the Tarim Basin, the main idea is that the wetter climate in the northwestern China inland corresponding to low eccentricity. The authors provide new records of proxy parameters of both chemical elements and a magnetic susceptibility index. Generally this MS was well organized. But the present version is not acceptable, my comment is major revision.

Specific comments:

(1) This work was based on an earlier report by Heermance et al. (2018), who proposed eolian sand record exceeding 1000 m. As a basic knowledge, desert is source area of dust (erosion area), it is difficult to image an accumulation of thousand meters of dune sands. From this MS, I know the so-called dune sands are mostly alternations of coarse-sandstone beds and fine fluvial-lacustrine siltstones, otherwise, it is impossible to have thousand meters of dune sands! Nie et al. pointed out that they collected samples from part of the section of Heermance et al. (2018), so, it is necessary to have a figure to show how the new sample collections correspond to the stratigraphy reported by Heermance et al. (2018).

(2) The main idea of this MS is to discuss dry-wet climate. However, all the used indices of Si, Al, Fe, HIRM are sensitive to lithology variations between coarse sandstone (e.g., more quartz) and fine siltstone (e.g., more clay minerals), they are not easy link to dry-wet climatic cycles. To do this, you should carry out stable isotopic (carbon and oxygen) analyses, together with pollen results, to have convincing paleoclimatic records.

(3) As we know, there are many magnetic susceptibility parameters can be easily measured, why only HIRM was used?

(4) For Figure 1, I do not think this simple DEM map can show the increased moisture transport to the Tarim Basin.

(5) There are up to four Eccentricity minimums between 8.8 and 8.5 Ma without their corresponds of so called "disappearances of eolian dune strata", please see their Figure 2.

Reviewer #2:

Remarks to the Author:

Nie et al., report a record extracted from eolian strata interbedded with the fluvio-lacustrine deposits from northwestern Tarim Basin of China. It is interesting that the record indicates periodic wetting likely occurred at Earth's low orbital eccentricity during the late Miocene (from 10-7.7 Ma). Taking the modern climate condition as an analog and further inspecting using model simulation, they proposed that the enhanced westward moisture transport associated with weakened East Asian summer monsoon during the low eccentricity periods play an important role in prompting the occurrence of these periodic wetting, which is opposite to the wetting pattern observed in deserts at monsoon marginal regions. This proposition is unconventional and worth of exploration. However, the deficiency in the chronology, the reliability of record and the discussion hindering me to recommend the publication of this manuscript.

Firstly, for the chronology, although the section was dated by magnetostratigraphy again with relatively high resolution and is consistent with that obtained by Heermance et al (2018), I don't think this chronology is reliable, either the chronology established by Heermance et al. (2018), because there is no any radiogenic or fossil data exist to pin this section to GPTS, and in particular, this section comprises of eolian and fluvio-lacustrine deposits which are likely to be discontinuous deposition. Also, it might be inappropriate to use the linear sedimentation rate for the eolian and fluvio-lacustrine deposits. The sedimentary rates should be plotted along with the proxy indices series. It is important

for evaluating the reliability of the chronology.

Secondly, the key finding in this study is that the wetting periods were aligned with the low eccentricity and opposite to the East Asian summer monsoon records. There are several loess-paleosoil records spanning the late Miocene. Is there any monsoon record which could correlate or comparable with the record in this study (e.g, demonstrating dominant eccentricity cycles during the 10.0-7.7 Ma.)? Such comparison could further corroborate the findings in this study, provided that the present chronology is further confirmed and reliable.

Thirdly, figure 2 shows the time series all the proxy indices analyzed in this study. It seems that all these proxies are highly correlated with each other and related to the grain size changes. It is likely that sedimentary sorting contribute the changes of these proxies (I don't agree with the authors that there are no sedimentary sorting bias), although the changes in the sedimentary facies are also related to the climate and environment change. Since the authors take the present condition as an analog for the past, the present river system in Tarim Basin could be regarded as the river system in the past. The runoffs of the rivers in this Basin are dominated by the ice melting from and precipitation over the high mountains and the precipitation in the Basin is hardly to affect the runoff. The precipitation at the high mountain regions are more important than the precipitation within the Basin. I therefore urge the authors to do the back trajectory analysis for the precipitation events during the summer over the past 30 year for the surrounding high altitude region around the study site, and further backup the influence of easterly moisture transports on the precipitation at the study site.

Line 168-171 the authors claimed that "Climate simulation using late Miocene paleogeography and topography (ref. 36) suggest that it is more likely that the recent observed westward monsoonal moisture transport to the Tarim Basin would have occurred more frequently during the late Miocene (supplementary fig. 8)". I didn't see any discussion about the moisture transport to Tarim basin in this paper. It just focused on the influence of Tethys Sea shrinkage on the aridification of the Sahara. Maybe, I missed something in this paper or the authors just reanalyzed the model outputs in this paper. If so, more informations about precipitation, wind field, seasonality of this model in Asia should be presented in the supplementary materials.

For the supplementary Fig. 8, easterly monsoonal moisture transport along the north side of Tibetan Plateau can be seen, as exhibited by the arrows. However, such moisture transports can only reach to ~90E and the study site is located in ~77 E. Such large gap is hardly to be explained by the model resolution or uncertainties and should be taken into consideration.

Line 174-176. The citation is wrong here. In Ref 34, Zhang (2021) did not use late Miocene geological boundary conditions for model simulation. Reanalysis data were used in this study. Ref 36 also is not relevant.

Line 179-181 How can the authors conclude that monsoonal moisture was likely more easily transported to the study area during the late Miocene than that during the recent decades?

Reviewer #3:

Remarks to the Author:

In this manuscript the authors describe ~725 m of Miocene interbedded fluvial and eolian sediments in the Tarim Basin and, based on paleomagnetic stratigraphy, correlate the fluvial deposits to minima in Earth's orbital eccentricity values between ~10-7.7 Ma. The authors also used General Circulation Models for the Miocene and the modern analog to conclude the source of moisture for the fluvial deposits to be from the northeast and brought into the Tarim Basin by strong anti-cyclonal air flow

over Mongolia. This research is important as it seeks to identify the fundamental controls on climate oscillations on one of the few mid-latitude deserts on Earth.

I cannot assess the strength of the modelling, so I will focus on the sedimentary deposits and geochronology. I think that the magnetostratigraphy (with samples points every 0.5-2 m) looks quite good, and the correlation to the geomagnetic polarity timescale fairly straightforward. However, incredibly precise geochronology is key to test the authors' hypothesis that wetting events (i.e. fluvial deposits) correlate to periods of low eccentricity. Alternative hypotheses would be that wetting occurs during maxima in orbital eccentricity, as well as the hypothesis that wetting events occur during both minima and maxima in orbital eccentricity. Given the fact that there are not independent age controls, and that sedimentation rates varied, I do not think the data allows for the discrimination between these various hypotheses. When you look at Supplementary Figure 3, the gray bars show correlations between minima in eccentricity and high HIRM (fluvial deposits), but not all eccentricity peaks (low eccentricity) have gray bars and there are many HIRM peaks that do not correlate to lows in eccentricity. The authors also did 'tune' the data at several locations to move HIRM peaks from high eccentricity to low eccentricity. In the end, I just do not think the data has precise enough age control to allow the authors to correlate 'wetting events' to periods of low eccentricity.

Reviewer #4:

Remarks to the Author:

Through a high-precision age control of a strata in the northwestern Tarim Basin, China, along with coherent variations of environmental variables measured across the strata, Nie et al. found that the climate here went through periodic wettings and dryings during the Late Miocene. Most notably, they found the climate was wetter when the eccentricity of the Earth's orbit was lower, opposite to what was found for deserts at the monsoon margins. The observations seem to be robust and the interpretation reasonable. This study provides novel understanding of how hydrological cycle in a remote non-monsoonal region responds to orbital forcing. The manuscript is well written and deserves publication on N. Comms. In my opinion, if a few things, to be described below and mostly about the modeling part, can be clarified.

1. Since the authors claim that the region of interest is a non-monsoonal region, the annual precipitation is then not dominated by summer precipitation. Therefore, it is insufficient to just look at the changes in June-July-August (JJA) precipitation from either reanalysis data or the climate model simulations. They should demonstrate that the region is indeed wetter annually at low eccentricity than at high eccentricity.
2. The eccentricity has a long period during which ~5 cycles of precession occur. The authors have demonstrated that the precipitation (so far for JJA only) responds to the change of eccentricity under the present-day precession. It is well known that the precession has a strong influence on the monsoonal circulation and maybe also the precipitation over the northwestern Tarim Basin, and the present-day precession is at a phase such that the Asian summer monsoon is relatively weak. I think it is necessary that the authors also demonstrate the response of precipitation in the region of interest to eccentricity for other precessions. The precession during the mid-Holocene would be a good choice, during which the northern hemisphere received moderately more solar insolation in summer than at present. If the modeling shows that the response of precipitation in the northwestern Tarim Basin to eccentricity is similar to that under present-day precession or the response is statistically insignificant, then the conclusion made by the authors is better supported.
3. The authors used the simultaneous weakening of East Asian summer monsoon and increasing of precipitation over the northwestern Tarim Basin in recent decades as an analog for the changes during the Late Miocene, and then made an inference of the mechanism by which the precipitation responded to the eccentricity change. However, the weakening of East Asian summer monsoon during the past

decades was not due to orbital changes, but might be due to multiple reasons, such as atmospheric aerosol over East Asia, Atlantic multi-decadal oscillation or Pacific decadal oscillation. This should be pointed out and briefly discussed.

4. Statistical significance test should be done for Fig. 3c, as done for Fig. S7.

5. L60, "eccentricit" is a typo.

6. L97, maybe "coherent" is better than "similar" here?

Reviewer #1 (Remarks to the Author):

General comment:

This manuscript (MS) aims to discuss the orbital forcing of dry-wet cycle in the Tarim Basin, the main idea is that the wetter climate in the northwestern China inland corresponding to low eccentricity. The authors provide new records of proxy parameters of both chemical elements and a magnetic susceptibility index. Generally this MS was well organized. But the present version is not acceptable, my comment is major revision.

Specific comments:

(1) This work was based on an earlier report by Heermance et al. (2018), who proposed eolian sand record exceeding 1000 m. As a basic knowledge, desert is source area of dust (erosion area), it is difficult to image an accumulation of thousand meters of dune sands. From this MS, I know the so-called dune sands are mostly alternations of coarse-sandstone beds and fine fluvial-lacustrine siltstones, otherwise, it is impossible to have thousand meters of dune sands! Nie et al. pointed out that they collected samples from part of the section of Heermance et al. (2018), so, it is necessary to have a figure to show how the new sample collections correspond to the stratigraphy reported by Heermance et al. (2018).

Thanks. A figure showing how the interval where we collected samples corresponds to the original stratigraphy in Heermance et al. (2018)¹ is added as Supplementary Fig. 1. In addition, the eolian member of the Wuqia group reported in Heermance et al. (2018) is defined as the section between 525-1698 m. This study focuses on only ~700 m of this section, where the dune strata are prominent and well-exposed.

(2) The main idea of this MS is to discuss dry-wet climate. However, all the used indices of Si, Al, Fe, HIRM are sensitive to lithology variations between coarse sandstone (e.g., more quartz) and fine siltstone (e.g., more clay minerals), they are not easy link to dry-wet climatic cycles. To do this, you should carry out stable isotopic (carbon and oxygen) analyses, together with pollen results, to have convincing paleoclimatic records.

Sorry for the confusion. This paper aims to recognize patterns of disappearance of the sand dune strata (environmental wetting) and the underlying forcing mechanisms. Therefore, the used environmental variation parameters are powerful to achieve this goal by converting depositional environmental variations to quantitative records. So, in this case, the correlation between lithofacies and environmental proxies does not jeopardize the validity of our inference of environmental wet-dry variations.

That said, we did measure stable isotope data for 193 samples from dune and interdune strata and present thin section evidence supporting carbonates are not detrital. The stable isotope data show consistent pattern as the other parameters (Fig. 2), providing further support to our conclusions. We tried pollen analysis for a few samples but did not find any pollen, unfortunately. This is likely because pollen grains have been oxidized in this arid environment.

(3) As we know, there are many magnetic susceptibility parameters can be easily measured, why only HIRM was used?

We now include additional magnetic parameter data and they are similar to HIRM variations (Supplementary Fig. 2).

(4) For Figure 1, I do not think this simple DEM map can show the increased moisture transport to the Tarim Basin.

Reply: Yes, we only use this as a cartoon elaboration. Fig. 3 is the one to examine moisture transport because this figure is based on model simulations.

(5) There are up to four Eccentricity minimums between 8.8 and 8.5 Ma without their corresponds of so called “disappearances of eolian dune strata”, please see their Figure 2.

Reply: Thanks for pointing out this pattern. In response to this comment, we added these sentences in the revised version (lines 188-201):

The interval of 8.8-8.5 Ma has three eccentricity lows but our paleoenvironmental data do not show clear changes. We notice that this interval has low sediment accumulation rates (Supplementary Fig. 3). We suspect that our equal interval sampling strategy may have skipped the interdune layers because interdune layers may be thinner over this interval than the intervals with high sediment accumulation rates. Alternatively, this interval may include hiatuses caused by erosion of sand dunes, which have removed interdune layers. Despite these possibilities, a careful examination of the grain size data (Figs. 2, Supplementary Figs. 7 and 8) shows three intervals of grain size decrease and silt proportion increase between 8.8 and 8.5 Ma, aligning well with low eccentricity. Furthermore, content of nanometer-scale ferrimagnetic grains and hematite increased over the low eccentricity intervals of 8.8-8.5 Ma, as can be told from increased ARM susceptibility and HIRM values (Supplementary Fig. 7), indicating enhanced weathering associated with environmental wetting. This pattern is consistent with the observed wetting aligning with low eccentricity pattern in the other intervals.

Reviewer #2 (Remarks to the Author):

Nie et al., report a record extracted from eolian strata interbedded with the fluvio-lacustrine deposits from northwestern Tarim Basin of China. It is interesting that the record indicates periodic wetting likely occurred at Earth's low orbital eccentricity during the late Miocene (from 10-7.7 Ma). Taking the modern climate condition as an analog and further inspecting using model simulation, they proposed that the enhanced westward moisture transport associated with weakened East Asian summer monsoon during the low eccentricity periods play an important role in prompting the occurrence of these periodic wetting, which is opposite to the wetting pattern observed in deserts at monsoon marginal regions. This proposition is unconventional and worth of exploration. However, the deficiency in the chronology, the reliability of record and the discussion hindering me to recommend the publication of this manuscript.

Firstly, for the chronology, although the section was dated by magnetostratigraphy again with relatively high resolution and is consistent with that obtained by Heermance et al (2018), I don't think this chronology is reliable, either the chronology established by Heermance et al. (2018), because there is no any radiogenic or fossil data exist to pin this section to GPTS, and in particular, this section comprises of eolian and fluvio-lacustrine deposits which are likely to be discontinuous deposition. Also, it might be inappropriate to use the linear sedimentation rate for the eolian and fluvio-lacustrine deposits. The sedimentary rates should be plotted along with the proxy indices series. It is important for evaluating the reliability of the chronology.

Reply: The Heermance et al. (2018)¹ age model is consistent with Qiao et al. (2016)² and the section is consistent with suggested Miocene-Pleistocene age in the geological map of this area (Fig. R1). Qiao et al. (2016)² and Heermance et al. (2018)¹ provide two, independently collected and analyzed, >3000m-thick sections only 20-km apart. The magnetostratigraphy is remarkably consistent between these two studies, and is unambiguously correlated with the late Miocene-Pliocene. In particular, the portion between 11-2.5 Ma has an almost perfect correlation with the Geomagnetic Polarity Time Scale (GPTS). Combined with the geological map, the magnetostratigraphy is unique and robust. Unlike other sections having debate on the age model, this section does not have alternative interpretation of the age model. Moreover, this age model for the Xiyu, Artushi, and Wuqia data are consistent with published age models in correlative strata in the Kashgar basin 100 km to the west^{3,4}. Therefore, we feel that the age model of Qiao et al. (2016)², Heermance et al. (2018)¹, and our own is robust.

We agree that eolian section of the strata may suffer from erosion. To address this critique, we examined if there are interdune siltstone layers (indicating wetting) aligning with geomagnetic reversals, which would give unambiguous phasing correlation between siltstone layer occurrence and eccentricity. We found two interdune siltstone layers near 9.3 Ma and 9.7 Ma (Fig. 2) occurred over geomagnetic reversals. So, we can use these two layers to pin down phasing relationship between environmental wetting and eccentricity variations. The observation that each of these two interdune layers corresponds to one eccentricity minimum supporting our argument: wetting corresponding to eccentricity minimum, which is also supported by our model simulations.

We also examined our paleomagnetic data in more detail and deleted those that do not have consistent declination and inclination. After excluding these samples, we found three reversals loosely constrained, with samples recording opposite polarities are 75, 78, and 125 kyr apart (marked using blue triangles in Fig. 2). Therefore, the part of the age model based on these reversals have large uncertainties and should not be used to infer phasing relationship between siltstone layers and eccentricity. Eight reversals are constrained tightly (less than

20 kyr; marked with red triangles), and siltstone layers constrained by these reversals aligns well with eccentricity minima (Fig. 2), providing further support to our model.

We note that below ~50 m (>9.85 Ma), siltstone appearance aligns with high eccentricity, suggesting a different pattern. However, this interval and the interval below this interval in Heermance et al. (2018)¹ are dominated by siltstones, with sand dune layers only appearing briefly, a pattern opposite to the portion younger than 9.85 Ma. This pattern suggests that the environments before 9.85 Ma in the Tarim Basin was generally wetter than after 9.85 Ma. Therefore, the cyclic brief wetting pattern after 9.85 Ma should not be applied to the earlier strata in which environmental wetting is more common than environmental drying.

We also plotted the SAR versus data as suggested (Supplementary Fig. 3). The plot shows that the interval ~8.8-8.3 Ma has longest duration of low sediment accumulation rate. We consider hiatuses may exist due to erosion of sand dune movements, as suggested by the reviewer. We discussed this possibility in lines 192-196. We suspect that existence of hiatuses may explain the lack of clear correlation between grain size and eccentricity over 8.8-8.5 Ma.

In summary, we believe that the careful examination of the paleomagnetic data proves the robustness of our observations, and we want to thank the reviewer for the thoughtful comments.

Figure R1. The location of the West Kepintagh (WK) section in the regional geological map, modified from Turner et al., (2010)⁵. The section span in Heermance et al. (2018)¹ is marked by the red bar inside the pink rectangle.

Secondly, the key finding in this study is that the wetting periods were aligned with the low eccentricity and opposite to the East Asian summer monsoon records. There are several loess-paleosol records spanning the late Miocene. Is there any monsoon record which could correlate or comparable with the record in this study (e.g, demonstrating dominant eccentricity cycles during the 10.0-7.7 Ma.)? Such comparison could further corroborate the findings in this study, provided that the present chronology is further confirmed and reliable.

Reply: This is an excellent suggestion! There are two loess-paleosol sections available covering the studied interval in the western Chinese Loess Plateau (Qinan section⁶ and Zhuanglang core⁷). However, their magnetic susceptibility records are not consistent with each other (Fig. R2). Nie et al. (2017)⁸ generated magnetic precipitation records of the late Miocene from the HTTL section of the eastern Qaidam Basin and the trend compared well with the magnetic susceptibility record from Qinan loess⁶. Nie et al. (2017)⁸ argued that the HTTL precipitation was controlled by Asian monsoon. Interestingly, the HTTL precipitation records from the eastern Qaidam Basin shows strong 100-kyr cycles (Supplementary Fig. 9), similar to the Tarim records here. And the wetting in eastern Qaidam Basin is in phase with high eccentricity, contrary to the Tarim pattern over the same interval (Supplementary Fig. 9). Therefore, these analyses support our hypothesis.

We have clarified that the general validity of the age model for the studied section above and discussed the limitation of the upper portion age model.

Figure R2. A comparison of the magnetic susceptibility record between Qinan⁶ and Zhuanglang⁷ sites on the western Chinese Loess Plateau, and magnetic precipitation record from the HTTL section of the eastern Qaidam Basin.

Thirdly, figure 2 shows the time series all the proxy indices analyzed in this study. It seems that all these proxies are highly correlated with each other and related to the

grain size changes. It is likely that sedimentary sorting contribute the changes of these proxies (I don't agree with the authors that there are no sedimentary sorting bias), although the changes in the sedimentary facies are also related to the climate and environment change. Since the authors take the present condition as an analog for the past, the present river system in Tarim Basin could be regarded as the river system in the past. The runoffs of the rivers in this Basin are dominated by the ice melting from and precipitation over the high mountains and the precipitation in the Basin is hardly to affect the runoff. The precipitation at the high mountain regions are more important than the precipitation within the Basin. I therefore urge the authors to do the back trajectory analysis for the precipitation events during the summer over the past 30 year for the surrounding high altitude region around the study site, and further backup the influence of easterly moisture transports on the precipitation at the study site.

Reply: We agree with the reviewer that sedimentary facies variations are also related to climate and environmental variations. Therefore, using the parameters potentially suffering from sediment sorting issue is indeed not a problem even if they are suffering from sorting issue.

We did the suggested back trajectory analysis, and compared the result with real satellite data and flow field analysis. However, the back trajectory analysis is not consistent with flow fields (Figures R3–R7) and satellite images (Figure R8), suggesting that back trajectory may not be able to recover moisture path faithfully in this area. For example, back trajectory analysis suggests that the water vapour was mainly from the northwest for a heavy precipitation event in Wensu (close to the study site) by the westerlies from July 28 to 31, 2018 (Figure R3). However, flow fields (Figures R3–R7) and satellite images (Figure R8) indicate that the water vapour was mainly transported by the monsoon (southerly and easterly moisture transports) rather than the westerlies towards Wensu during these days. Therefore, we do not show the back trajectories in the revised manuscript.

That said, we do agree with the comment of the reviewer indicating that precipitation at the high mountain regions is important for moisture changes in the Tarim Basin. Supplementary Fig. 11 suggests that precipitation changes over the surrounding high mountains are more significant than that over the Tarim Basin between the easterly and westerly moisture transport events. Therefore, precipitation changes over the surrounding high mountains are also closely related to the easterly moisture transport.

Figure R3. Moisture sources for a heavy precipitation event in Wensu from July 28 to 31, 2018, based on backward trajectory analysis using NOAA HYSPLIT model.

Figure R4. Mean fluxes of water vapour ($\text{kg}\cdot\text{m}^{-1}\cdot\text{s}^{-1}$) in the middle–upper troposphere (700–100 hPa, a), and in the lower troposphere (surface to 700 hPa, b) on July 28, 2018.

Figure R5. Mean fluxes of water vapour ($\text{kg}\cdot\text{m}^{-1}\cdot\text{s}^{-1}$) in the middle–upper troposphere (700–100 hPa, a), and in the lower troposphere (surface to 700 hPa, b) on July 29, 2018.

Figure R6. Mean fluxes of water vapour ($\text{kg}\cdot\text{m}^{-1}\cdot\text{s}^{-1}$) in the middle-upper troposphere (700–100 hPa, a), and in the lower troposphere (surface to 700 hPa, b) on July 30, 2018.

Figure R7. Mean fluxes of water vapour ($\text{kg}\cdot\text{m}^{-1}\cdot\text{s}^{-1}$) in the middle–upper troposphere (700–100 hPa, a), and in the lower troposphere (surface to 700 hPa, b) on July 31, 2018.

Figure R8. Satellite imagery of the movement of clouds from July 28 to 31, 2018

(a–d).

Line 168-171 the authors claimed that “Climate simulation using late Miocene paleogeography and topography (ref. 36) suggest that it is more likely that the recent observed westward monsoonal moisture transport to the Tarim Basin would have occurred more frequently during the late Miocene (supplementary fig. 8)”. I didn’t see any discussion about the moisture transport to Tarim basin in this paper. It just focused on the influence of Tethys Sea shrinkage on the aridification of the Sahara. Maybe, I missed something in this paper or the authors just reanalyzed the model outputs in this paper. If so, more information about precipitation, wind field, seasonality of this model in Asia should be presented in the supplementary materials.

We clarified this point in lines 232-239.

For the supplementary Fig. 8, easterly monsoonal moisture transport along the north side of Tibetan Plateau can be seen, as exhibited by the arrows. However, such moisture transports can only reach to ~90E and the study site is located in ~77 E. Such large gap is hardly to be explained by the model resolution or uncertainties and should be taken into consideration.

The moisture transport in Figure S8 was calculated using the simulation results conducted by the low-resolution (~3.75°) version of the NorESM by Zhang et al. (2014)⁹ (Reference 36 in the initial submission). Our simulation results, using high-resolution (~1°) version of the CAM4, shows a clear easterly monsoonal moisture transport along the north side of Tibetan Plateau towards the study site at 10 Ma (Figure 3a). We removed Figure S8 in the revised manuscript to avoid confusion.

Line 174-176. The citation is wrong here. In Ref 34, Zhang (2021) did not use late Miocene geological boundary conditions for model simulation. Reanalysis data were used in this study. Ref 36 also is not relevant.

It should be Zhang et al. (2014)⁹ ref 42 in the revised manuscript.

Line 179-181 How can the authors conclude that monsoonal moisture was likely more easily transported to the study area during the late Miocene than that during the recent decades?

Monsoonal moisture transport in northwestern China are only synoptic scale events which cannot be detected in the summer mean water vapor flux field during the recent decades and pre-industrial (Figures 3b, Supplementary Fig. 10a). In contrast, monsoonal moisture transport prevailed in northwestern China throughout the whole summer during the late Miocene. Therefore, we conclude that monsoonal moisture was likely more easily transported to the study area during the late Miocene than that during the recent decades.

We have re-discussed this part in lines 232-239 because we removed Figure S8 in the revised manuscript.

Reviewer #3 (Remarks to the Author):

In this manuscript the authors describe ~725 m of Miocene interbedded fluvial and eolian sediments in the Tarim Basin and, based on paleomagnetic stratigraphy, correlate the fluvial deposits to minima in Earth's orbital eccentricity values between ~10-7.7 Ma. The authors also used General Circulation Models for the Miocene and the modern analog to conclude the source of moisture for the fluvial deposits to be from the northeast and brought into the Tarim Basin by strong anti-cyclonal air flow over Mongolia. This research is important as it seeks to identify the fundamental controls on climate oscillations on one of the few mid-latitude deserts on Earth.

I cannot assess the strength of the modelling, so I will focus on the sedimentary deposits and geochronology. I think that the magnetostratigraphy (with samples points every 0.5-2 m) looks quite good, and the correlation to the geomagnetic polarity timescale fairly straightforward. However, incredibly precise geochronology is key to test the authors' hypothesis that wetting events (i.e. fluvial deposits) correlate to periods of low eccentricity. Alternative hypotheses would be that wetting occurs during maxima in orbital eccentricity, as well as the hypothesis that wetting events occur during both minima and maxima in orbital eccentricity. Given the fact that there are not independent age controls, and that sedimentation rates varied, I do not think the data allows for the discrimination between these various hypotheses. When you look at Supplementary Figure 3, the gray bars show correlations between minima in eccentricity and high HIRM (fluvial deposits), but not all eccentricity peaks (low eccentricity) have gray bars and there are many HIRM peaks that do not correlate to lows in eccentricity. The authors also did 'tune' the data at several locations to move HIRM peaks from high eccentricity to low eccentricity. In the end, I just do not think the data has precise enough age control to allow the authors to correlate 'wetting events' to periods of low eccentricity.

We thank the reviewer for the critical and thorough comments. To address this critique, we focus on two interdune strata (indicating wetting) near 9.3 Ma and 9.7 Ma (Fig. 2), where two geomagnetic reversals align with each of them. So, one can conclude that no or negligible amount of erosion has occurred at these two locations. The observation that these two strata correspond to eccentricity minima provides solid evidence supporting our argument-wetting corresponding to eccentricity minimum, which is also supported by our model simulations.

Furthermore, we rechecked the uncertainties of the age model at different part of the section. We recognized three loosely constrained reversals and the age model of the strata constraining by these reversals have large uncertainties. Eight reversals are tightly constrained in this section, and we observe good alignment between siltstone layers and eccentricity minima for strata constraining by these well-defined reversals. This careful examination of the age tie points used to build the age model provides further support to our hypothesis regarding relationship between desert wetting and eccentricity.

We do note that the pattern with eccentricity older than 9.8 Ma may not be consistent with the younger portion, because the older interval has large portion of wet intervals, different from the young interval (Fig. R9).

Figure R9. Plot showing the bottom of the eolian dune-containing sequence is dominated by wet environment and upper sequence is dominated by dry environment¹. The boundary is near 9.8 Ma.

Reviewer #4 (Remarks to the Author):

Through a high-precision age control of a strata in the northwestern Tarim Basin, China, along with coherent variations of environmental variables measured across the strata, Nie et al. found that the climate here went through periodic wettings and dryings during the Late Miocene. Most notably, they found the climate was wetter when the eccentricity of the Earth's orbit was lower, opposite to what was found for deserts at the monsoon margins. The observations seem to be robust and the interpretation reasonable. This study provides novel understanding of how hydrological cycle in a remote non-monsoonal region responds to orbital forcing. The manuscript is well written and deserves publication on N. Comms. in my opinion, if a few things, to be described below and mostly about the modeling part, can be clarified.

1. Since the authors claim that the region of interest is a non-monsoonal region, the annual precipitation is then not dominated by summer precipitation. Therefore, it is insufficient to just look at the changes in June-July-August (JJA) precipitation from either reanalysis data or the climate model simulations. They should demonstrate that the region is indeed wetter annually at low eccentricity than at high eccentricity.

We added new simulations which demonstrate that at annual scale this region is wetter at lower eccentricity (Supplementary Fig. 12).

2. The eccentricity has a long period during which ~5 cycles of precession occur. The authors have demonstrated that the precipitation (so far for JJA only) responds to the change of eccentricity under the present-day precession. It is well known that the precession has a strong influence on the monsoonal circulation and maybe also the precipitation over the northwestern Tarim Basin, and the present-day precession is at a phase such that the Asian summer monsoon is relatively weak. I think it is necessary that the authors also demonstrate the response of precipitation in the region of interest

to eccentricity for other precessions. The precession during the mid-Holocene would be a good choice, during which the northern hemisphere received moderately more solar insolation in summer than at present. If the modeling show that the response of precipitation in the northwestern Tarim Basin to eccentricity is similar to that under present-day precession or the response is statistically insignificant, then the conclusion made by the authors is better supported.

We would like to discuss with the reviewer about this suggestion. In our model, precipitation increase in the Tarim basin is accompanied with enhanced monsoonal moisture input into the Tarim basin caused by the weakened East Asian summer monsoon. For 6 ka, East Asian summer monsoon circulation is strong as is documented by many lakes and desert region records^{10,11}, leading to scarce monsoonal moisture input into the Tarim basin. In this situation, our model does not require precipitation increase in the Tarim Basin because monsoonal moisture has been precipitated in East Asia. So we disagree that at the other precessional setting the Tarim Basin would be wetter as well in low eccentricity than high eccentricity.

We performed the recommended simulations (Fig. R10), and did not observe increased precipitation in the Tarim Basin, and we explained our reasoning above.

Figure R10. Simulated annual precipitation (shaded, mm day⁻¹) when using 6k

precession. a, Results using 10 Ma background boundary conditions, 0.05 eccentricity and 6ka precession. b, Results using 10 Ma background boundary conditions, 0.01672 eccentricity and 6ka precession. c, Precipitation differences when increasing eccentricity from 0.01672 (year 1950) to 0.05 when using 6k precession. Dotted regions indicate differences that are significant at the 0.05 confidence level.

3. The authors used the simultaneous weakening of East Asian summer monsoon and increasing of precipitation over the northwestern Tarim Basin in recent decades as an analog for the changes during the Late Miocene, and then made an inference of the mechanism by which the precipitation responded to the eccentricity change. However, the weakening of East Asian summer monsoon during the past decades was not due to orbital changes, but might be due to multiple reasons, such as atmospheric aerosol over East Asia, Atlantic multi-decadal oscillation or Pacific decadal oscillation. This should be pointed out and briefly discussed.

We added these texts to incorporate the suggestions of the reviewer: “although the driving forces of the EASM may differ between the orbital and decadal timescales (ref 43-45).” in lines 246-247.

4. Statistical significance test should be done for Fig. 3c, as done for Fig. S7.

Done.

5. L60, “eccentricit” is a typo.

Done

6. L97, maybe “coherent” is better than “similar” here?

Done.

References

1. Heermance, R. V. et al. Erg deposition and development of the ancestral Taklimakan Desert (western China) between 12.2 and 7.0 Ma. *Geology* **46**, 919-922 (2018).
2. Qiao, Q. Q., Huang, B. C., Piper, J. D. A., Deng, T. & Liu, C. Y. Neogene magnetostratigraphy and rock magnetic study of the Kashi Depression, NW China: Implications to neotectonics in the SW Tianshan Mountains. *J. Geophys. Res.* **121**, 1280-1296 (2016).
3. Heermance, R. V., Chen, J., Burbank, D. W. & Wang, C. S. Chronology and tectonic controls of Late Tertiary deposition in the southwestern Tian Shan foreland, NW China. *Basin Res.* **19**, 599-632 (2007).
4. Chen, J. et al. Magnetostratigraphy of the Upper Cenozoic strata in the Southwestern Chinese Tian Shan: rates of Pleistocene folding and thrusting. *Earth Planet. Sci. Lett.* **195**, 113-130 (2002).
5. Turner, S. A., Cosgrove, J. W. & Liu, J. G. Controls on lateral structural variability along the Keping Shan Thrust Belt, SW Tien Shan Foreland, China. *Geol. Soc. Spec. Publ.* **348**, 71-85 (2010).
6. Guo, Z. T. et al. Onset of Asian desertification by 22 Myr ago inferred from loess deposits in China. *Nature* **416**, 159-163 (2002).
7. Qiang, X. K. et al. New eolian red clay sequence on the western Chinese Loess Plateau linked to onset of Asian desertification about 25 Ma ago. *Sci.*

China Earth Sci. **54**, 136-144 (2010).

8. Nie, J. S. et al. Dominant 100,000-year precipitation cyclicality in a late Miocene lake from northeast Tibet. *Sci. Adv.* **3**, e1600762 (2017).
9. Zhang, Z. S. et al. Aridification of the Sahara desert caused by Tethys Sea shrinkage during the Late Miocene. *Nature* **513**, 401-404 (2014).
10. Lu, H. Y. et al. Late Quaternary aeolian activity in the Mu Us and Otindag dune fields (north China) and lagged response to insolation forcing. *Geophys. Res. Lett.* **32**, L21716 (2005).
11. Chen, F. H. et al. Holocene moisture evolution in arid central Asia and its out-of-phase relationship with Asian monsoon history. *Quat. Sci. Rev.* **27**, 351-364 (2008).

Reviewers' Comments:

Reviewer #1:

Remarks to the Author:

The authors have replied my comments, I have no further suggestions.

Reviewer #2:

Remarks to the Author:

The revised manuscript by Nie et al. is an improvement of the previous version. However, the authors did not satisfy the concerns raised by the referees. There are still several issues need to be clarified before it can be accepted.

There is still no solid evidence to reinforce the present chronology and improve it for arguing about the eccentricity is dominant factor affecting the wetting of the Tarim Basin during the middle Miocene. Particularly, if we consider that about 10% strata are covered and there is no data from these missing portions. I understand that failing to collect some data is expected in the field investigation of geological sections. However, as seen from the figure 2, there are about 35 meters are covered at the lower part of the section (~125m in thickness), such large gap might cause significant uncertainty in the magnetostratigraph and weaken the arguments in the manuscript.

The authors tried using the interdune siltstone layers aligned with geomagnetic reversals to prove the phase relationship between environmental wetting and eccentricity lows. Two layers near 9.3 Ma and 9.7 Ma have been identified to be fitted with the proposition. However, there is still a layer not fitted with such proposition, i.e., the layer near 8.25 Ma, where the geomagnetic reversal was well constrained.

The kernel point of this study is that the weak East Asian summer monsoon caused by low eccentricity may lead to the moisture transports to the west, associated with anticyclonic circulation over Mongolia . Following this logic, the precessional variation of East Asian summer monsoon will also induce the changes in the moisture transport to the west China, such changes might be more significant than the changes caused by the eccentricity changes. In fact, the difference between the model simulation of 6 ka and present (at both high and low eccentricity) would be a good test on this assumption. Further analysis on the model simulation in this study could help to clarify this question.

Supposing that the low eccentricity may cause increased precipitation at Tarim Basin during the late Miocene, the relatively low eccentricity during the periods of eccentricity lows 1-2, 5-6, 18-19 and 22-23, as shown in figure 2, would generate four wetting periods lasting about two eccentricity cycles. However, we did see no persisting wetting period in this record. The author need to explore this issue and explain this reasonably.

Reviewer #4:

Remarks to the Author:

The modeling results presented by the authors look reasonable to me, at least for this particular model. It is also interesting to know from their new results that the wetting and drying of Tarim desert is not sensitive to eccentricity when the precession is configured such that the summer insolation of the Northern Hemisphere is high. This provides a testing point for the future when the temporal resolution of the records is high enough to resolve the precession timescale clearly and robustly. I have no further comments.

REVIEWER COMMENTS

Reviewer #1 (Remarks to the Author):

The authors have replied my comments, I have no further suggestions.

Reply: We thank the reviewer for the constructive comments, which improved the manuscript significantly.

Reviewer #2 (Remarks to the Author):

The revised manuscript by Nie et al. is an improvement of the previous version. However, the authors did not satisfy the concerns raised by the referees. There are still several issues need to be clarified before it can be accepted.

Reply: We thank the reviewer for the positive and encouraging comments and for careful examination of the manuscript and the response letter. The comments made by the reviewers were very constructive and we appreciate these insightful comments. The other two reviewers have approved our revised manuscript. Below, we further address the several points raised by reviewer 2.

There is still no solid evidence to reinforce the present chronology and improve it for arguing about the eccentricity is dominant factor affecting the wetting of the Tarim Basin during the middle Miocene. Particularly, if we consider that about 10% strata are covered and there is no data from these missing portions. I understand that failing to collect some data is expected in the field investigation of geological sections. However, as seen from the figure 2, there are about 35 meters are covered at the lower part of the section (~125m in thickness), such large gap might cause significant uncertainty in the magnetostratigraph and weaken the arguments in the manuscript.

Reply: We appreciate the concern of the reviewer. We added these texts to address the comment of the reviewer in lines 116-122: There are three covered intervals in the studied section (~10% in thickness), but the upper two covered intervals do not align with geomagnetic reversals (Fig. 2), so their existence does not increase uncertainties of our age model. The lower covered interval does overlap with one geomagnetic reversal (Fig. 2), but there are five tightly constrained reversals (<20 kyr) from 10-9.3 Ma for the lower part of the section, so the existence of this covered interval only has limited impact on the precision of our age model.

The authors tried using the interdune siltstone layers aligned with geomagnetic reversals to prove the phase relationship between environmental wetting and eccentricity lows. Two layers near 9.3 Ma and 9.7 Ma have been identified to be fitted with the proposition. However, there is still a layer not fitted with such proposition, i.e., the layer near 8.25 Ma, where the geomagnetic reversal was well constrained.

Reply: We thank the reviewer for the careful examination of the figure. We zoomed in and examined the reversal near 8.25 Ma after seeing the comments of the reviewer. It turns out that this reversal aligns with an environmental drying interval (Fig. R1).

Furthermore, the dominant wet interval during 8.21-8.25 Ma (highlighted by the gray bar in Fig. R1), as can be told by finer grain size, aligns with low eccentricity, consistent with our model. Therefore, considering this reversal reinforces our conclusions. Indeed, although we present evidence suggesting eccentricity minimum correspond to and caused interdune siltstone appearance in Tarim strata, we did not want to convey the message that all wetting intervals correspond to eccentricity minima because other mechanisms may also exist. However, demonstrating Tarim desert wetting and greening mainly linked to eccentricity minimum and Asian summer monsoon weakening during the late Miocene is useful to understand future dry-wet variations in Asian inland area, and this is the main theme of this manuscript. This link is clear even considering the reversal near 8.25 Ma (Fig. R1). We clarify this point in the revised version (lines 303-305) and we thank the reviewer for the careful observation: Although eccentricity minimum and Asian monsoon weakening may not be the exclusive reason for Tarim desert wetting and greening during the late Miocene, the data suggest that this is the major mechanism.

Figure R1. Comparison of the (a) lithology, (b) median grain size and (c) eccentricity during 8.29-8.15 Ma. Red triangle represents the paleomagnetic age control point.

The kernel point of this study is that the weak East Asian summer monsoon caused by low eccentricity may lead to the moisture transports to the west, associated with anticyclonic circulation over Mongolia. Following this logic, the precessional variation of East Asian summer monsoon will also induce the changes in the moisture transport to the west China, such changes might be more significant than the changes caused by the eccentricity changes. In fact, the difference between the model simulation of 6 ka and present (at both high and low eccentricity) would be a good test on this assumption. Further analysis on the model simulation in this study could help to clarify this question.

Reply: This is a great point and we performed the recommended exercises.

In the low eccentricity setting (Fig. R2a), the precipitation variations in the Tarim Basin and the NE Tibetan Plateau/Chinese Loess Plateau do not show sensitive responses to

precessional variations. This is reasonable because the forcing (insolation) has lower amplitude when eccentricity is low.

In the high eccentricity setting (Fig. R2b), the precipitation variations in the Tarim Basin and eastern China (referred here to roughly 25-50° N; 95-120° E; including the NE Tibetan Plateau/Chinese Loess Plateau) show sensitive responses to precessional variations, and these two areas show opposite precipitation pattern, just like that in Fig. 3c. This confirms the inference of the reviewer. It also explains why the Tarim Basin was wetter in low eccentricity intervals than high eccentricity intervals (Fig. 2), which can be attributed to moisture penetration to the Tarim basin only at a portion of the high eccentricity intervals controlled by precessional variations. In contrast, under low eccentricity setting (corresponding to less eastern China precipitation associated with weaker East Asian summer monsoon), moisture could have been more commonly transported to the Tarim Basin because Tarim precipitation was not sensitive to precessional value variations in this eccentricity setting.

We note that our simulations show a weaker East Asian summer monsoon precipitation in eastern China under 6 ka precession setting than that under present precession setting, with the late Miocene boundary conditions. This is consistent with the results of Marzocchi et al. (2015)⁴. And Xu et al. (2020) attributed decreased precipitation in eastern China to the westward shift of the western Pacific Subtropical High⁵, which was also detected in our simulation under 6 ka precession setting, shown by anomalous anticyclonic moisture transport in eastern China (Fig. R2b). These studies are consistent and complimentary to our simulations.

We added these contents in lines 274-297.

Fig. R2 Simulated late Miocene summer precipitation (shaded, mm day⁻¹) differences between 6 ka and 0 ka precessional setting (6ka-0ka) under low (a) and high (b) eccentricity. The precipitation in Fig. R2b is similar to that in Fig. 3c. The experiment settings are outlined in Supplementary Table 2. Eccentricity settings are the same as in Fig. 3: Low eccentricity = 0.01672 (year 1950); high eccentricity = 0.05. Dotted regions indicate differences that are significant at the 90% confidence level. Squares indicate location of the Tarim Basin, NE Tibetan Plateau (NE TP), and the Chinese Loess Plateau (CLP).

Supposing that the low eccentricity may cause increased precipitation at Tarim Basin during the late Miocene, the relatively low eccentricity during the periods of

eccentricity lows 1-2, 5-6, 18-19 and 22-23, as shown in figure 2, would generate four wetting periods lasting about two eccentricity cycles. However, we did see no persisting wetting period in this record. The author need to explore this issue and explain this reasonably.

Thanks for pointing out this potential issue. Although eccentricity was generally low over the intervals corresponding to eccentricity 1-2, 5-6, 18-19, and 22-23 (Fig. 2g), it does exist fluctuations. For example, the Tarim Basin was wetter at intervals corresponding to eccentricity minimum 18 and 19 defined in Fig. 2 (Fig. R3); by contrast, for the time interval between eccentricity minimum 18 and 19, the Tarim Basin experienced less persistent wetting (Fig. R3). We suspect that the desert wetting only occurred at eccentricity minimum intervals instead of all low eccentricity intervals. That is, only at intervals of most persistent weaker East Asian summer monsoon modulated by eccentricity minimum, moisture could be more commonly and persistently brought from East Asia to inland Asia to cause Tarim desert wetting during the late Miocene.

Fig. R3 Eccentricity, precessional index and Fe content variations in the Tarim Basin over the interval ~9.4-9.3 Ma

Reviewer #4 (Remarks to the Author):

The modeling results presented by the authors look reasonable to me, at least for this particular model. It is also interesting to know from their new results that the wetting and drying of Tarim desert is not sensitive to eccentricity when the precession is configured such that the summer insolation of the Northern Hemisphere is high. This provides a testing point for the future when the temporal resolution of the records is high enough to resolve the precession timescale clearly and robustly. I have no further comments.

Reply: We thank the reviewer for the constructive comments, which improved our manuscript significantly.

References

1. Heermance, R. V. et al. Erg deposition and development of the ancestral Taklimakan Desert (western China) between 12.2 and 7.0 Ma. *Geology* **46**, 919-922 (2018).
2. Qiao, Q. Q., Huang, B. C., Piper, J. D. A., Deng, T. & Liu, C. Y. Neogene magnetostratigraphy and rock magnetic study of the Kashi Depression, NW China: Implications to neotectonics in the SW Tianshan Mountains. *J. Geophys. Res.- Sol. Ea.* **121**, 1280-1296 (2016).
3. Turner, S. A., Cosgrove, J. W. & Liu, J. G. Controls on lateral structural variability along the Keping Shan Thrust Belt, SW Tien Shan Foreland, China. *Geol. Soc. Spec. Publ.* **348**, 71-85 (2010).
4. Marzocchi, A. et al. Orbital control on late Miocene climate and the North African monsoon: insight from an ensemble of sub-precessional simulations. *Clim. Past* **11**, 1271-1295 (2015).
5. Xu, H. et al. Juxtaposition of Western Pacific Subtropical High on Asian Summer Monsoon Shapes Subtropical East Asian Precipitation. *Geophys. Res. Lett.* **47**, e2019GL084705 (2020).

Reviewers' Comments:

Reviewer #2:

Remarks to the Author:

Nie et al. have largely clarified my concerns. It is very interesting that the East Asian summer monsoon precipitation decreased in East China as the summer insolation increased and Indian summer monsoon precipitation increased. Such scenario seems incompatible with the observations from the geological records, and might be caused by the different boundary condition. What about the Asian monsoon intensity? Can we state that "Asian monsoon weakening" for this scenario? The paper by Xu et al. attributed decreased precipitation in southwest China but not eastern China to the westward shift of the western Pacific Subtropical High. It is unsuitable to take this study as an analogy to explain the simulation result with the late Miocene boundary conditions.

Also, double check the supplementary fig6. There is no much difference in panel a and panel c, while panel b and panel d show apparent differences. Panel c looks like a photo of sandstone, but not the siltstone.

Nie et al. have largely clarified my concerns. It is very interesting that the East Asian summer monsoon precipitation decreased in East China as the summer insolation increased and Indian summer monsoon precipitation increased. Such scenario seems incompatible with the observations from the geological records, and might be caused by the different boundary condition. What about the Asian monsoon intensity? Can we state that "Asian monsoon weakening" for this scenario? The paper by Xu et al. attributed decreased precipitation in southwest China but not eastern China to the westward shift of the western Pacific Subtropical High. It is unsuitable to take this study as an analogy to explain the simulation result with the late Miocene boundary conditions.

Reply: We agree with the reviewer and changed "Asian monsoon weakening" to "East Asian monsoon weakening" in the title. This way, the problem raised by the reviewer is not an issue any more.

We have replaced Xu et al. (2020) with another article (Dai et al., 2021) to explain the simulation result. Dai et al. (2021) used the westward shift of the western Pacific Subtropical High to explain decreased precipitation in eastern China during the mid-Holocene and other periods with high summer insolation, consistent with our case.

Reference:

Dai, G. W. et al. A modeling study of the tripole pattern of East China precipitation over the past 425 ka. *J. Geophys. Res-Atmos.* **126**, e2020JD033513 (2021).

Also, double check the supplementary fig6. There is no much difference in panel a and panel c, while panel b and panel d show apparent differences. Panel c looks like a photo of sandstone, but not the siltstone.

Reply: We double-checked supplementary figure 6 and found that we did use a wrong picture in panel c unintentionally. We thank the reviewer for finding this error. The wrong picture is now replaced by the right one, which clearly shows that the sample at 86 m is a siltstone.